# Ameliorative Effect of Imperatorin on *Dermatophagoides pteronyssinus*-Induced Allergic Asthma by Suppressing the Th2 Response in Mice

**DOI:** 10.3390/molecules27207028

**Published:** 2022-10-18

**Authors:** Chia-Chen Hsieh, Yan-Yan Ng, Wei-Sung Li, Chung-Yuh Tzeng, Tsai-Yi Hsu, Wan-Hsiang Huang, Jen-Chieh Tsai

**Affiliations:** 1Department of Medicine Division of Chest Medicine, Cheng Ching Hospital, Taichung 40764, Taiwan; 2Department of Medicinal Botanicals and Foods on Health Applications, Da-Yeh University, ChangHua 51591, Taiwan; 3Department of Pediatrics, Chung-Kang Branch, Cheng Ching Hospital, Taichung 40764, Taiwan; 4Plant Pathology Division, Taiwan Agricultural Research Institute, Council of Agriculture, Executive Yuan, Wufeng 41362, Taiwan; 5Department of Orthopedics, Taichung Veterans General Hospital, Taichung 40705, Taiwan; 6Institute of Biomedical Sciences, National Chung Hsing University, Taichung 40227, Taiwan; 7Department of Bioresources, Da-Yeh University, Changhua 51591, Taiwan

**Keywords:** imperatorin, asthma, *Dermatophagoides pteronyssinus*, Th2 response

## Abstract

Imperatorin is a furanocoumarin derivative and an effective ingredient in several Chinese medicinal herbs. It has favorable expectorant, analgesic, and anti-inflammatory effects. In this study, we investigated whether imperatorin has protective effects against *Dermatophagoides pteronyssinus* (*Der p*)-induced asthma in mice. Lung and bronchial tissues were histopathologically examined through hematoxylin–eosin staining. The concentrations of immunoglobin E (IgE), IgG1, IgG2a in serum and those of T helper 1 (Th1) and two cytokines and eosinophil-activated chemokines in bronchoalveolar lavage fluid (BALF) were detected using an enzyme immunoassay. Histological examination revealed that imperatorin reduced inflammatory cell infiltration, mucus hypersecretion, and endothelial cell hyperplasia. The examination also indicated that imperatorin could reduce the inflammatory cell count in BALF as well as IgE and IgG1 expression in serum, but IgG2a expression was significantly increased. Imperatorin reduced the production of interleukin (IL)-4, IL-5, and IL-13 by Th2, promoted the production of interferon-γ and IL-12 by Th1, and increased the production of IL-10 in bronchoalveolar lavage fluid. These findings suggest that imperatorin has a considerable anti-inflammatory effect on *Der p*-induced allergic asthma in mice.

## 1. Introduction

Allergic asthma is a chronic inflammatory respiratory condition, and its symptoms include reversible airway inflammation, airway reorganization, respiratory tract hyperresponsiveness, and respiratory tract obstruction [1]. The respiratory systems of patients with asthma tend to be irritated by environmental factors that induce such symptoms as chest pain and coughing. Moreover, long-term respiratory tract infection may trigger mucus swelling, increased mucus secretion, and respiratory tract narrowing, causing breathing difficulties. Several types of asthma allergens, such as mold, pollen, animal dander, and house dust mites, have been identified. These allergens may cause respiratory tract inflammation and asthma attacks [2]. In Taiwan, allergic asthma is primarily caused by dust mites [3]. Allergic asthma typically involves eosinophilic airway inflammation because mucus is overproduced and bronchial hyperresponsiveness causes the airways to narrow. Dust mite allergens (*Dermatophagoides pteronyssinus*, *Der p*) are associated with the activity of cysteine proteases that can cleave key molecules that constitute the tight junctions of the epithelium. Moreover, cysteine proteases render epithelial cells more fragile and exposed to allergen activity [4].

Macrophages or dendritic cells activate allergens, causing T cells to secrete interleukin (IL)-4, IL-5, and IL-13 to initiate a T helper cell 2 (Th 2) immune response [5]. IL-4 regulates allergen-specific B lymphocytes to trigger the production of immunoglobin E (IgE), which may cause the migration of inflammatory cells in the region of the allergen. Thus, asthma has traditionally been categorized as an adaptive immune disease in which lymphocytes overreact to antigens and trigger a Th2 immune response, subsequently causing an increase in mast cells, basophils, and eosinophils [2]. Mast cells release the cytokines IL-13 and IL-5. IL-13 promotes the production of mucus in the respiratory tract and activates the secretion of eotaxins to accumulate eosinophils. IL-5 maintains eosinophil survival in tissues and stimulates the release of a large amount of toxic substances, which destroy the respiratory epithelial cells and cause tracheal damage [6]. To repair injury to the respiratory tract, the parenchymal tissue of the respiratory tract undergoes regeneration [7]. Connective tissue replaces damaged airway tissue, and airway reorganization causes airway changes, including goblet cell and myofibroblastic hyperplasia, epithelial cell hypertrophy and fibrosis, increased mucus secretion, and bronchial wall thickening [8,9]. Eventually, the respiratory tract is narrowed, blocked, and becomes more sensitive, resulting in chronic symptoms.

Imperatorin is a derivative of furanocoumarin and exists in the form of long thin white needles or crystals, which are easily soluble in nonpolar solvents and insoluble in water (Figure 1). Imperatorin is widely present in several Chinese medicinal herbs, including *Angelica dahurica*, *Radix glehniae*, *A. sinensis*, *Rhodiola rosea*, *Bupleurum*
*c**hinese*, and *Foeniculum vulgare*. It exhibits several pharmacological properties, including anticancer, anti-inflammatory, antioxidant, neuroprotective, and immunomodulatory effects, but it might have hepatotoxicity in mice at higher doses (40 mg/kg) [10]. However, no study has been conducted on its effects on *Der p*-induced allergic asthma. Accordingly, the purpose of this study was to explore whether imperatorin can alleviate allergic asthma caused by dust mite allergen. We used *Der p* to induce sensitization in mice and administered imperatorin to the mice to evaluate whether the substance could alleviate the inflammatory response caused by *Der p*-induced allergic asthma.

## 2. Results

### 2.1. Effects of Imperatorin on Der p-Specific IgE, IgG1, and IgG2a in Serum

The levels of IgE and IgG1 in serum increased and the level of IgG2a in serum decreased in the sensitized group compared with the control group. However, the IgE and IgG1 in the mice treated with 10 mg/kg of imperatorin decreased significantly (*p* < 0.05; Figure 2a,b), but the IgG2a in the mice treated with 5 and 10 mg/kg of imperatorin increased significantly (*p* < 0.05; Figure 2c).

### 2.2. Histopathological Analyses of Lungs and Trachea

Histological examination of the lungs revealed that the lung structure of the control group was maintained. In the mice with *Der p*-induced allergic asthma, the lung tissues were inflamed, and cellular infiltration had occurred. Moreover, *Der p* led to the recruitment of eosinophils and lymphocytes around the perivascular and peribronchial spaces. Macrophage aggregation in the alveolar space and bronchial epithelial hyperplasia and goblet cell modification in the bronchus were also observed. In contrast to the *Der p*-induced group, the imperatorin-treated group (10 mg/kg) exhibited decreased inflammatory cell infiltration in the lungs (Figure 3, Table 1). In addition, the results revealed that the tracheal tissue exposed to *Der p* exhibited submucosal eosinophil infiltration, whereas that treated with imperatorin exhibited decreased eosinophil infiltration (Figure 4, Table 1). These results indicated that imperatorin can inhibit inflammatory infiltration in *Der p*-induced asthmatic mice.

### 2.3. Effects of Imperatorin on Inflammatory Cell Count in Bronchoalveolar Lavage Fluid (BALF)

To understand whether imperatorin ameliorated *Der p*-induced inflammation, we quantified the inflammatory cells in bronchoalveolar lavage fluid (BALF). The results shown in Figure 5 reveal that the sensitized mice had a higher inflammatory cell count compared with the control group; by contrast, the mice treated with imperatorin (5 mg/kg and 10 mg/kg) had significantly reduced numbers of cells. However, there was no significant difference in the treatment with imperatorin at dose of 1 mg/kg compared with the *Der p*-induced group.

### 2.4. Effects of Imperatorin on Cytokines in BALF of Der p-Induced Allergic Asthma Mice

The imbalance of Th1/Th2 cytokines is a property of asthma attacks. Therefore, determining the concentration of Th1/Th2 cytokines in BALF can enable the ascertainment of whether imperatorin can regulate such an imbalance. Compared with *Der p*–treated mice, in imperatorin-treated mice, IL-4 (10 mg; Figure 6a), IL-13 (5 and 10 mg/kg; Figure 6b), and IL-5 (5 and 10 mg/kg; Figure 6c) concentrations decreased but IL-10 (5 and 10 mg/kg; Figure 6d), IL-12 (10 mg/kg; Figure 6e), and interferon-γ (IFN-γ) (1, 5, and 10 mg/kg; Figure 6f) concentrations increased with the imperatorin treatment.

## 3. Discussion

Asthma is caused by long-term contact with allergens and is an airway inflammation syndrome involving various inflammatory mediators [11]. Previous studies have asserted that inflammatory mediators in the microenvironment are critical factors in regulating the differentiation of primitive T cells, which may be related to the sensitization process [12,13]. At present, corticosteroids such as dexamethasone are the main treatment for asthma and inflammation. However, the side effects of steroids have caused restrictions on use [14]. Therefore, it is necessary to find new drugs, and it is a trend to find effective ingredients from natural products. Oral ingestion of imperatorin may affect the immune systems of mice through different pathways. The previous study indicated that imperatorin might have hepatotoxicity in mice (40 mg/kg, 28 days) [10], and this might be related to the dose used in the experiment. Therefore, the dose selection in this study could also consider toxicity factors, so the safer dose 10 mg/kg was used as the highest dose.

Imperatorin can stimulate antigen-presenting cells to produce Th1 cytokines, thereby participating in immune system regulation [15,16,17]. We therefore investigated whether imperatorin ameliorates airway inflammation response in dust-mite-induced asthma. Compared with the sensitized group, the BALF of imperatorin-treated mice exhibited decreased total inflammatory cell count. Inflammatory cell infiltration, including eosinophil and lymphocyte accumulation, in the perivascular and bronchial spaces causes bronchial epithelial hyperplasia and goblet cell modification [18]. Histopathological analysis revealed that treatment with imperatorin effectively inhibited the infiltration of inflammatory cells, including eosinophils. Taken together, these findings indicate that imperatorin exerts anti-inflammatory effects on *Der p*-induced asthma.

Asthma is related to an imbalance of Th1 and Th2 cell responses followed by increases in various immune cells and inflammatory mediators [19]. IgE plays a key role in allergic asthma pathogenesis, and a high IgE antibody content has been observed after exposure to the dust mite allergen among patients with asthma [20,21]. We observed that after being sensitized with dust mites, the mice exhibited large amounts of IgE and IgG1 antibodies related to the Th2 immune response. Through the provision of imperatorin, the production of allergy-related antibodies, such as IgE and IgG1, were reduced. These findings support those of other studies which indicated imperatorin may inhibit the degranulation of IgE-mediated mast cells and the histamine release [17,22]. Conversely, the Th1-related specificity for IgG2a increased, particularly in the mice treated with imperatorin. These results revealed that imperatorin balances Th1 and Th2 cytokines in mice with *Der p*-induced allergic asthma.

IgG2a administered partially through the nasal cavity to mice with allergy-induced inflammation was demonstrated to reduce the total white blood cell count and goblet cell deformation in BALF and to stimulate the production of a large amount of Th1 cytokines; moreover, the anti-allergen specificity for IgG2a provided protection against allergy-induced respiratory tract inflammation [23]. IFN-γ produced by Th1 in BALF increased significantly in imperatorin-treated mice. Previous studies have suggested that imperatorin increases the production of IFN-γ in the lungs; hence, the Th1 immune response is activated to reduce the Th2 inflammatory response [15,16]. In this study, the mice treated with imperatorin produced large amounts of the Th1 cytokine IFN-γ after their stimulation with the dust mite allergens. IFN-γ is necessary to promote IgG2a production in B cells [24,25]. Therefore, imperatorin may have activated the Th1 cytokine IFN-γ, and thereby increased the dust mite specificity for IgG2a to provide effective anti-allergen activity.

Th2 cells initiate a cascade of allergic responses by activating interleukins, primarily IL-4 and IL-5, which results in the production of IgE, recruitment of eosinophils to the airways, and promotion of the growth of mucosal-type mast cells [26]. These cytokine-inducing signals lead to hyperresponsiveness of the respiratory tract and increased mucus secretions, triggering the symptoms of asthma. The Th2 cytokine IL-4 is essential for the production of Th2 cells. Th2 cells stimulate airway epithelial cells to secrete mucus. Through B cell class switching, IgE production can also be promoted [27]. In addition, airway eosinophils are activated by IL-5 to promote degranulation, and airway hyperresponsiveness, inflammation, and remodeling are affected by IL-13 [28]. Furthermore, the Th1 cytokines IFN-γ and IL-12 participate in the anti-Th2 cell response and IgE synthesis, thereby inhibiting asthma progression [29]. IFN-γ and IL-12 inhibit the production of IgE, and IgE is the cause of allergic reactions. Our results indicated that in imperatorin-treated mice, the Th2 cytokines IL-4, IL-5, and IL-13 decreased and the Th1 cytokines IFN-γ and IL-12 increased. These findings are consistent with that of a previous study reporting that the imperatorin can reduce the levels of IL-4 and IL-13 and furthermore inhibit the promotion of IgE [22]. IL-10 regulates inflammatory responses, which can inhibit monocytes/macrophages and the production of many proinflammatory cytokines [30]. Our results indicated that IL-10 increased with imperatorin treatment. Taken together, the findings indicate that imperatorin upregulates Th1 cytokines, downregulates allergy-related Th2 inflammation, and reduces the inflammatory response, including the allergen-induced Th2 immune response.

## 4. Materials and Methods

### 4.1. Materials

*Der p* powder was obtained from Allergon AB (Angelholm, Sweden). *Der p* powder was extracted using diethyl ether and then the filtrate was evaporated. PBS was added in to dissolve and transfer to dialysis membrane (cut off point: 12–14 KDa), and then it was dialyzed with water. The dialysate was freeze-dried to powder to obtain *Der p* extract. The extract was stored at −80 °C before use. Imperatorin was purchased from Sigma-Aldrich (St. Louis, MO, USA. Freund’s incomplete adjuvant was purchased from Difco (Detroit, MI, USA).

### 4.2. Animals

We assembled forty 6–8-week-old male BALB/c mice. The mice were purchased from the National Laboratory Animal Center and maintained in the Animal Center of Da-Yeh University. The breeding room was air-conditioned, with the temperature maintained at 22 °C ± 1 °C and relative humidity maintained at 55% ± 5%; a 12 h dark–light cycle was set (08:00 lights on, 20:00 lights off), and unlimited food and water resources were provided. The animal care and handling protocols were approved by the Animal Use Committee of Da-Yeh University (approval number: 106035).

### 4.3. Dust Mite Allergen Mouse Model

The BALB/c mice were randomly separated into five groups of eight. The first group (control group) and second group (*Der p* group) were administered normal drinking water, and the other three groups were orally administered imperatorin (1, 5, and 10 mg/kg in 0.5% carboxymethylcellulose separately) 2 weeks before the dust mite allergens were administered and then every day until the mice were diagnosed with asthma (4 weeks in total). The doses were chosen according to previous studies [31]. The investigator was blinded to the treatment groups.

The allergic sensitization and challenge protocol was in accordance with that used in our previous study [32]. Except for the control group, all groups were sensitized with the dust mite allergens *Der p* on day 0 and day 7 and subcutaneously injected at the base of the tail with an emulsion (50 μL) containing *Der p* (50 μg) in incomplete Freund’s adjuvant (Difco, Detroit, MI, USA). The control group was injected with 50 μL of normal saline water. On day 14, asthma was triggered by the dust mite allergens in the trachea under tiletamine/zolazepam (*Zoletil* 50, Virbac Corporation, Carros, France) anesthesia (20 mg/kg body weight), and the mice were subjected to treatment using an intranasal instillation of 50 μL of *Der p* antigen solution (1.0 mg/mL). The mice were maintained in an upright position for 1 min until the restoration of normal breathing. At 72 h after the last *Der p* challenge, the mice were sacrificed. The lung and trachea tissues were removed for histological analysis, and serums and BALF were collected (Figure 7).

### 4.4. Collection of Mouse Serum and BALF

Blood samples were allowed to rest for 1 h, followed by centrifugation at 2000× *g* (3000 rpm) for 5 min to precipitate the blood cells. The supernatant (serum) was collected and stored at −80 °C for further analysis of IgE, IgG1, and IgG2a expression.

The lungs were then lavaged using a syringe containing 1 mL of iced sterile saline solution; the lavage fluid was withdrawn and stored at low temperature. This process was repeated three times, and then the lavage fluid was collected [33]. The collected lavage fluid was centrifuged at 320× *g* (1200 rpm) for 10 min at 4 °C, and then the supernatant was stored at −80 °C. Each tube’s cell pellet was resuspended in phosphate-buffered saline (PBS), and a hemocytometer was used to measure the total inflammatory cell count.

### 4.5. Lung Histology

The lung and trachea of the mice were removed, and the required parts were dissected and maintained in 10% neutral buffered formalin. After gradual dehydration and paraffin embedding, the tissues were cut into 4–5 μm thick sections, fixed on glass slides, stained with hematoxylin–eosin, and observed under a light microscope to assess the damage level and histopathological changes.

### 4.6. Determination of Der p-Specific IgE, IgG1, and IgG2a in Serum

The expression levels of serum IgE, IgG1, IgG2a were quantified using an enzyme-linked immunosorbent assay (ELISA) kit in accordance with the manufacturer’s instructions (Invitrogen, Waltham, MA, USA).

### 4.7. Th1 and Th2 Cytokine Expression Detection in BALF

The content of various cytokines in BALF, including IL-4, IL-5, Il-10, IL-12, IL-13, and IFN-γ were determined using commercial mouse-specific ELISA kits (BioLegend, San Diego, CA, USA). Coating buffer with an appropriate amount of an anti-mouse cytokine antibody was added to a 96-well assay plate and then allowed to stand overnight at 4 °C. Subsequently, after being washed with PBS with Tween 20 (PBST) buffer, 200 μL/well of blocking buffer was added to the plate. After 2 h of reaction at room temperature, the plate was again washed with PBST buffer; then, 100 μL of sample was added. After another 2 h of reaction at room temperature, 100 μL/well of adequately concentrated anti-cytokine secondary antibodies was added. After another 2 h of reaction at room temperature, 100 μL/well of streptavidin-HRP was added and allowed to react for 20 min. Subsequently, tetramethylbenzidine was added and allowed to react for 20 min. Finally, 50 μL of (NH_4_)_2_SO_4_ was added to terminate the color reaction and determine the absorbance at 450 nm by a Synergy HT Multi-Mode Microplate Reader (BioTek, Winooski, VT, USA).

### 4.8. Statistical Analysis

All experimental data are presented as the mean ± standard error of the mean (SEM). Statistical analyses were performed with SPSS software for Windows, version 25 (IBM, Chicago, IL, USA). Data were statistically analyzed using one-way analysis of variance (ANOVA); the sensitized group and groups treated with different imperatorin concentrations were tested using Scheffe’s multiple range test. Histopathological analyses were performed by the non-parametric Kruskal–Wallis test followed by the Mann–Whitney U-test. Statistical significance was set at *p* < 0.05.

## 5. Conclusions

We discovered that imperatorin promoted the secretion of Th1 cytokines and reduced the Th2 cytokine response caused by dust-mite-induced allergies by inhibiting the IgE and IgG1 levels and increasing the production of IgG2a antibodies, which can diminish the allergic response. Moreover, imperatorin reduced lung inflammatory cell infiltration, increased IL-12 and IFN-γ production by Th1, reduced IL-4, IL-5, and IL-13 production by Th2, and increased IL-10 production in BALF. Our findings imply that imperatorin can offer protection against allergic asthma.

## Figures and Tables

**Figure 1 molecules-27-07028-f001:**
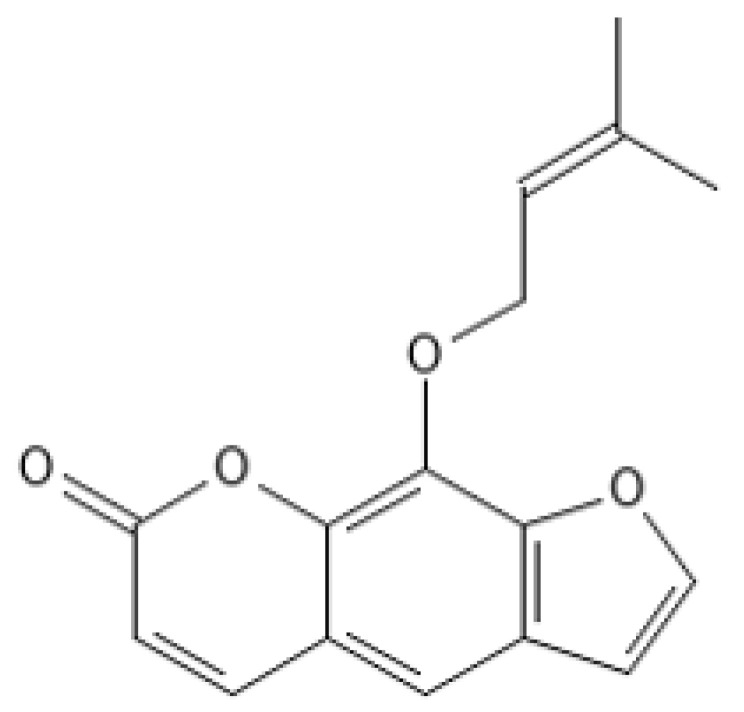
Chemical structure of imperatorin.

**Figure 2 molecules-27-07028-f002:**
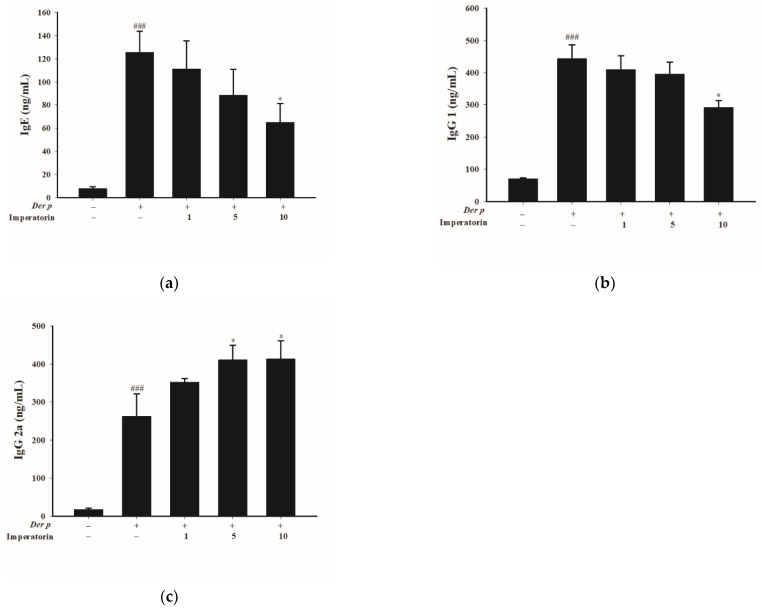
Effects of imperatorin on IgE, IgG1, and IgG2a expression in serum of mice with *Der p*-induced asthma. ### *p* < 0.001 compared with the control group, * *p* < 0.05 compared with *Der p*-induced group, *n* = 8 (one-way ANOVA followed by Scheffe’s multiple range test). All data are presented as the mean ± SEM.

**Figure 3 molecules-27-07028-f003:**
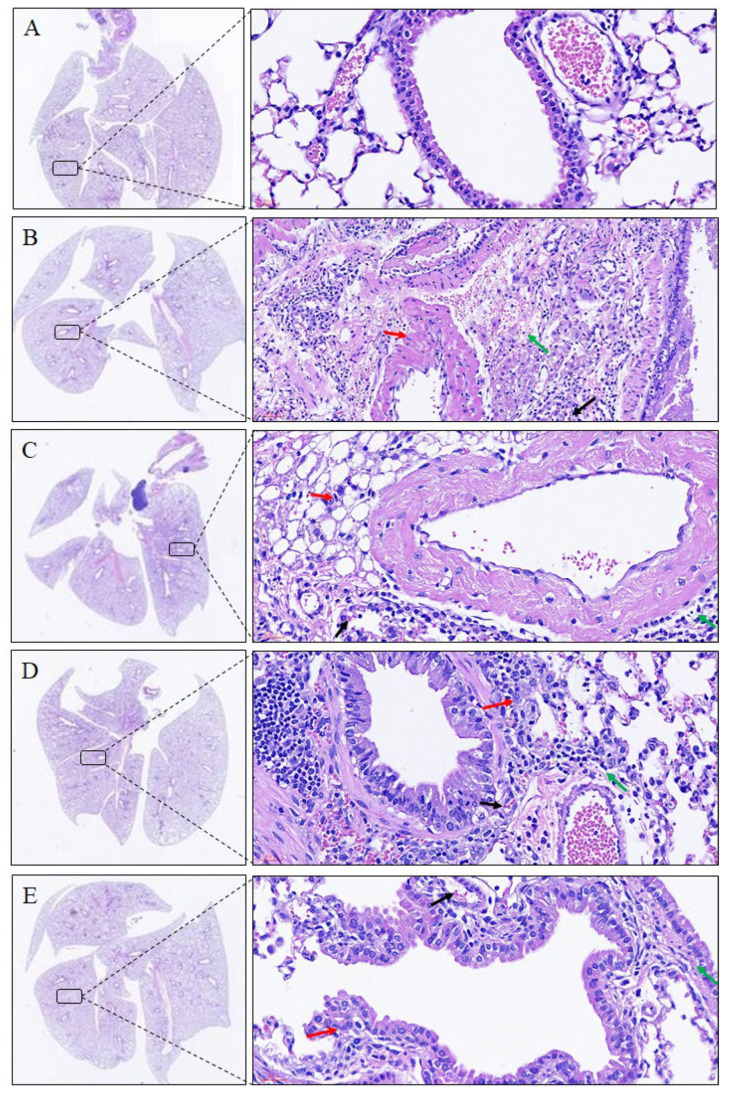
Lung histological examination results for *Der p*-induced allergic mice treated with imperatorin. Eosinophil is indicated by black arrow, macrophage is indicated by red arrow, and lymphocyte is indicated by green arrow. Hematoxylin–eosin staining results are presented at 400× magnification. (**A**) Control group; (**B**) *D**er p* group; (**C**) *D**er p* ± imperatorin 1 mg/kg group; (**D**) *D**er p* ± imperatorin 5 mg/kg group; (**E**) *D**er p* ± imperatorin 10 mg/kg group.

**Figure 4 molecules-27-07028-f004:**
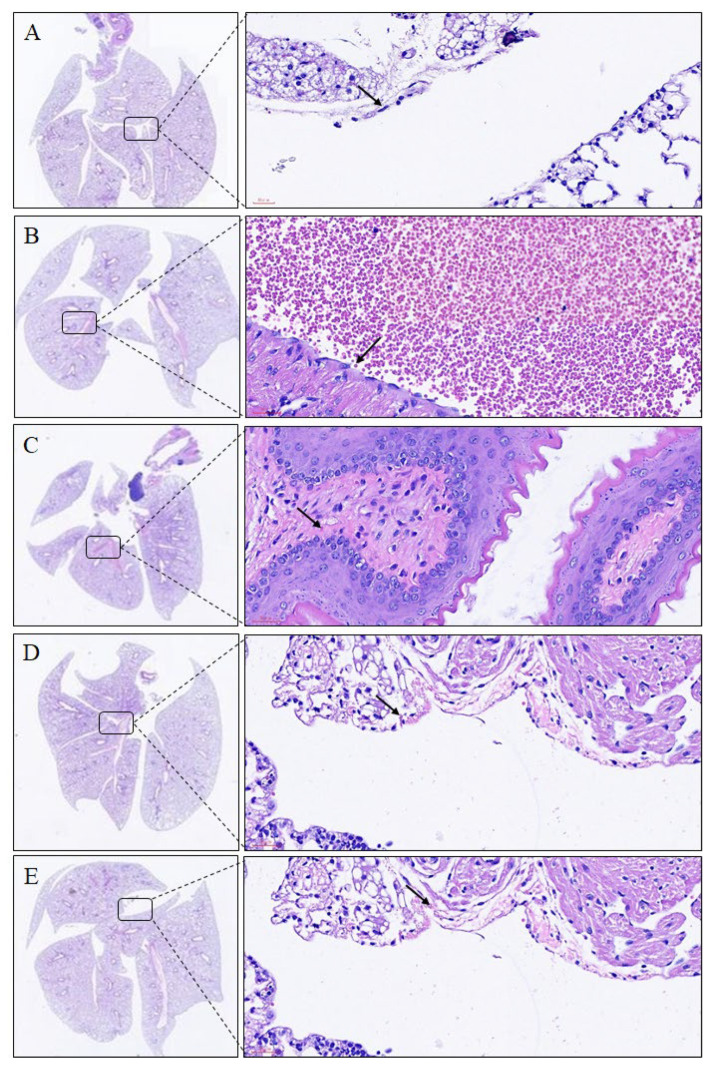
Trachea histological examination results for *Der p*-induced allergic mice treated with imperatorin. Eosinophil is indicated by black arrow, and hematoxylin–eosin staining results are presented at 400× magnification: (**A**) control group; (**B**) *Der p* group; (**C**) *Der p* ± imperatorin 1 mg/kg group; (**D**) *Der p* ± imperatorin 5 mg/kg group; (**E**) *Der p* ± imperatorin 10 mg/kg group.

**Figure 5 molecules-27-07028-f005:**
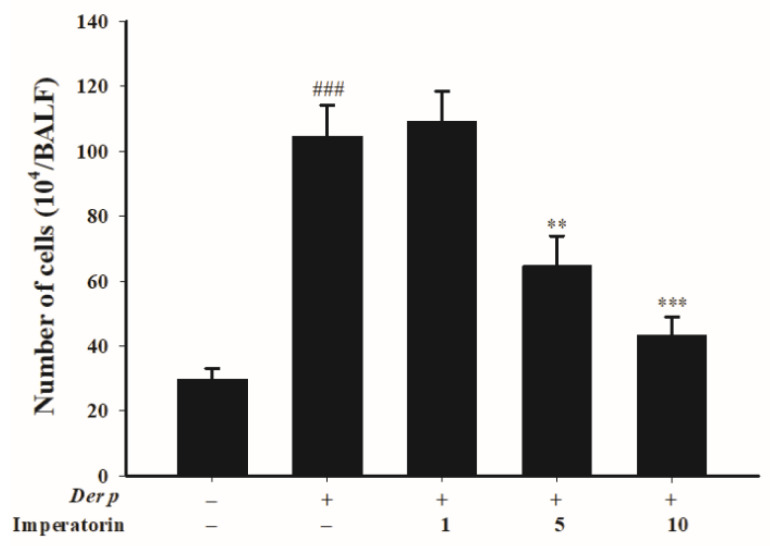
Total counts of inflammatory cells in the BALF of *Der p*-induced allergic mice treated with imperatorin. ### *p* < 0.001 compared with the control group, ** *p* < 0.01 and *** *p* < 0.001 compared with the *Der p*-induced group, *n* = 8 (one-way ANOVA followed by Scheffe’s multiple range test). All data are presented as the mean ± SEM.

**Figure 6 molecules-27-07028-f006:**
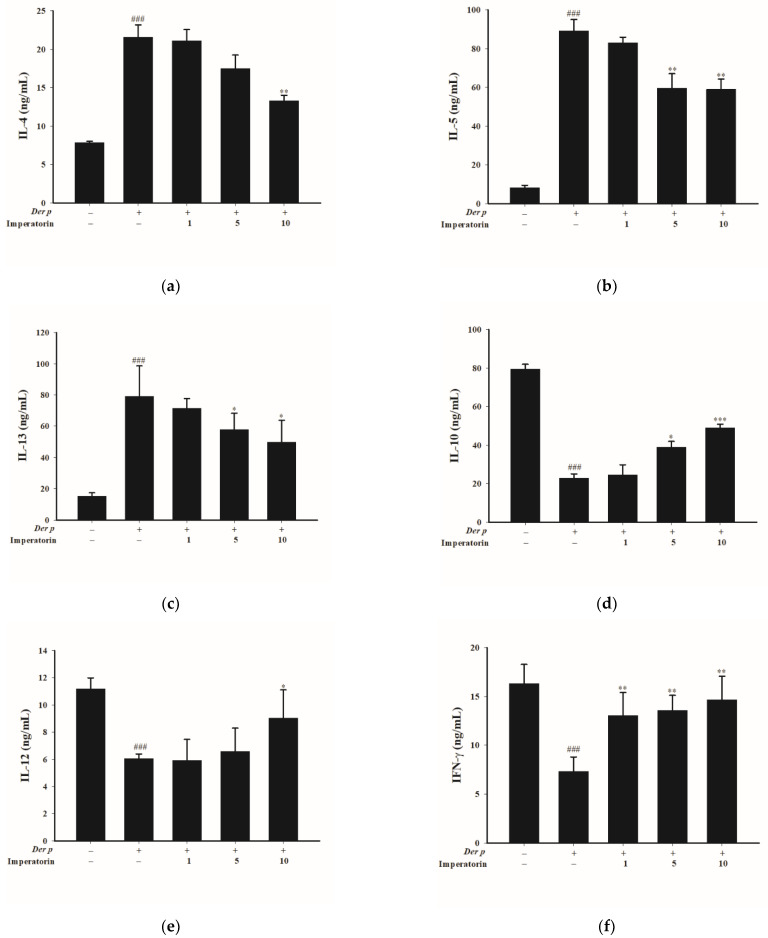
Effects of imperatorin on chemokine levels in the collected BALF of mice with *Der p*–induced allergic asthma. (**a**) IL-4, (**b**) IL-5, (**c**) IL-13, (**d**) IL-10, (**e**) IL-12, (**f**) IFN-γ. ### *p* < 0.001 compared with the control group, * *p* < 0.05, ** *p* < 0.01 and *** *p* < 0.001 compared with the *Der p*-induced group, *n* = 8 (one-way ANOVA followed by Scheffe’s multiple range test). All data are presented as the mean ± SEM.

**Figure 7 molecules-27-07028-f007:**
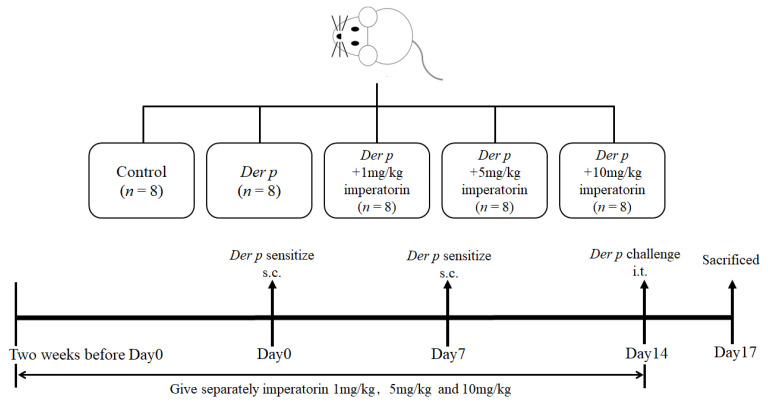
Schematic of the allergic sensitization and challenge protocol.

**Table 1 molecules-27-07028-t001:** Quantitative summary of the protective effect of imperatorin on *Der p*-induced allergic asthma based on histological observations.

Organ	Histopathological Findings	Groups ^1^				
		Control	*Der p*	*Der p*		
				Imperatorin (1 mg/kg)	Imperatorin (5 mg/kg)	Imperatorin (10 mg/kg)
Lung	Inflammation, eosinophilic and lymphocytic cells, perivascular and per bronchial, focal	0.0 ± 0.00	3.8 ± 0.43 ^#^	3.5 ± 0.50	2.5 ± 0.50	1.8 ± 0.43 *
	Aggregation, macrophage and giant cells, alveolar, focal	0.0 ± 0.00	3.5 ± 0.50 ^#^	3.3 ± 0.43	2.3 ± 0.43	1.5 ± 0.50 *
	Epithelial hyperplasia, bronchial, focal	0.0 ± 0.00	3.8 ± 0.43 ^#^	3.3± 0.83	2.5 ± 0.50 *	2.3 ± 0.43 *
	Mucification, goblet, bronchial, focal	0.0 ± 0.00	3.0 ± 0.00 ^#^	2.8 ± 0.43	2.3 ± 0.43 *	1.5 ± 0.50 *
Bronchial airways	Inflammation, submucosal focal	0.0 ± 0.00	3.6 ± 0.41 ^#^	3.8 ± 0.49	3.2 ± 0.43	2.5 ± 0.32 *

^1^ Lesions were graded from 1 to 5 depending on severity: 1 = minimal (<1%), 2 = slight (1–25%), 3 = moderate (26–50%), 4 = moderate/severe (51–75%), 5 = severe/high (76–100%). # Statistically significant difference between control and *Der p* groups at *p* < 0.05^.^ * Statistically significant difference between *Der p* and treated groups at *p* < 0.05.

## Data Availability

The data presented in this study are available in the article.

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
