# Peer review of "Ameliorative Effect of Imperatorin on Dermatophagoides pteronyssinus-Induced Allergic Asthma by Suppressing the Th2 Response in Mice"

_molecules, 2022, doi:10.3390/molecules27207028_

Round 1

Reviewer 1 Report

The manuscript entitled “Ameliorative Effect of Imperatorin on Dermatophagoides pteronyssinus–Induced Allergic Asthma by Suppressing Th2 Response in Mice” presents the evaluation of imperatorin protective effects against Dermatophagoides pteronyssinus (Der p)–induced asthma in mice.

The manuscript is well-written and very interesting, but in order to be published in Molecules journal, the authors are advised to make some corrections listed below:

1.      The authors are advised to use small letter for the second plant name species (Rhodiola rosea, Bupleurum chinese) – line 76

2.      The authors are advised to better explain the results from 2.3. section. Why for the 1mg/kg dose the results are different?

3.      The authors are advised to correct the term „coticosteroids” – line 164

4.      The authors are advised to add in the discussion part some information about the toxicity of imperatorin. As several studies showed, furanocoumarins have general toxicity. For imperatorin it was demonstrated the hepatotoxicity in mice (Deng M, Xie L, Zhong L, Liao Y, Liu L, Li X. Imperatorin: A review of its pharmacology, toxicity and pharmacokinetics. Eur J Pharmacol. 2020 Jul 15;879:173124. doi: 10.1016/j.ejphar.2020.173124. Epub 2020 Apr 24. PMID: 32339515.)  

Author Response

Comments and Suggestions for Authors

The manuscript entitled “Ameliorative Effect of Imperatorin on Dermatophagoides pteronyssinus–Induced Allergic Asthma by Suppressing Th2 Response in Mice” presents the evaluation of imperatorin protective effects against Dermatophagoides pteronyssinus (Der p)–induced asthma in mice.

The manuscript is well-written and very interesting, but in order to be published in Molecules journal, the authors are advised to make some corrections listed below:

  1. The authors are advised to use small letter for the second plant name species (Rhodiola rosea, Bupleurum chinese) – line 76

Ans: Thanks for your friendly reminder. The mistakes have been corrected.

  1. The authors are advised to better explain the results from 2.3. section. Why for the 1mg/kg dose the results are different?

Ans: Thanks for your friendly reminder. We have corrected the text as follow:

To understand whether imperatorin ameliorated Der p-induced inflammation, we quantified the inflammatory cells in bronchoalveolar lavage fluid (BALF). The results (Figure 5) revealed that the sensitized mice had a higher inflammatory cell count compared to the control group; by contrast, the mice treated with imperatorin (5 mg/kg and 10 mg/kg) had significantly reduced numbers of cells. However, there was no significant difference in the treatment with imperatorin at dose of 1 mg/kg compared with the Der p-induced group.

  1. The authors are advised to correct the term „coticosteroids” – line 164

Ans: Thanks for your friendly reminder. The mistake has been corrected.

  1. The authors are advised to add in the discussion part some information about the toxicity of imperatorin. As several studies showed, furanocoumarins have general toxicity. For imperatorin it was demonstrated the hepatotoxicity in mice (Deng M, Xie L, Zhong L, Liao Y, Liu L, Li X. Imperatorin: A review of its pharmacology, toxicity and pharmacokinetics. Eur J Pharmacol. 2020 Jul 15;879:173124. doi: 10.1016/j.ejphar.2020.173124. Epub 2020 Apr 24. PMID: 32339515.)  

Ans: Thanks for your friendly reminder. The toxicity and the dose selection of imperatorin were revised as follow:

The previous study indicated that imperatorin might have hepatotoxicity in mice (40 mg/kg, 28 days) [10], and this might be related to the dose used in the experiment. Therefore, the dose selection in this study could also consider toxicity factors, so the safer dose 10 mg/kg was used as the highest dose.

Reviewer 2 Report

In this work, the authors evaluate the effect of imperatorin in an experimental model of allergic asthma induced by a complete commercial extract of Der p in male BALB/c mice. The results show that oral administration of imperatorin at 10 mg/kg in Der p sensitized mice reduces the Th2 response and increases Th1 cytokine production with the production of IgG2a antibodies.

The work is interesting and other authors demonstrated the positive effect of imperatorina in lung injury, allergic responses and allergic rhinitis as it is cited in the discussion. However, there are some points that must be explained in more detail.

Materials and methods:

Line 224. Der p is a commercial allergenic extract or is it commercial raw material from which the authors have made the extract? Do you know the content in the main allergens of Der p? Is it LPS-free?

Line 251. The intranasal instillation was a total of 50 µg (50 µl at 1.0 mg/mL), did the mice resist this high concentration of allergens? In our experiments, using individualized allergens 5 µg/instillation is enough.

Line 257. Figure 5. On the line says “Der p sensitized two weeks ago” Is this correct? The text describes that the mice were sensitized by subcutaneous injection on day 0 and 7.

Line 278-279. Were the cytokines-ELISA used commercial or homemade? If they were commercial please, indicate.

Line 291. Statistical analysis.

Did the authors used any statistical software? Please, indicate.

The Scheffe’s test running after ANOVA is very flexible and with a low statistical power. It is better to use the Tukey’s test because it have a narrowed confidence interval and a high statistical power.

RESULTS.

Figure 3. Is the black arrow in the correct position?

Table 1. Do the authors have standardized measures of injury severity? How did the authors perform the statistical analysis in this case?

Figure 5. Why did the authors not used flow cytometry? It is usually less variable than cell count by hemocytometry and allows specific cell type counts such as eosinophils.

Figure 6. (d, e and f) Was the control group (-,-) include in the statistical study? Were the values of the control group (-,-) considered as the cytokine-basal values?

Discussion

Line 171-172. How did the authors count the macrophages?

Line 186 and 216. Ref. 15 and 20. The authors could compare these results in more detail.

Conclusions

 Line 299. “IgG1a” must be IgG1.

Introduction

Line 49-50. “In Taiwan, allergic asthma is primarily caused by dust mites”. The sentence needs a reference.

Line 62. Reference [2] appears after reference [3]. They are unordered.

Line 77. “Mill” would be in italic.

Line 77-78. The sentence needs references to support the beneficial effect of imperatorin. It would be convenient to indicate the existence of toxic effects of this compound (i. e. Sondhia et al., 2012).

Author Response

Comments and Suggestions for Authors

In this work, the authors evaluate the effect of imperatorin in an experimental model of allergic asthma induced by a complete commercial extract of Der p in male BALB/c mice. The results show that oral administration of imperatorin at 10 mg/kg in Der p sensitized mice reduces the Th2 response and increases Th1 cytokine production with the production of IgG2a antibodies.

The work is interesting ng and other authors demonstratthe positive effect of imperatorina in lung injury, allergic responses and allergic rhinitis as it is cited in the discussion. However, there are some points that must be explained in more detail.

Materials and methods:

Line 224. Der p is a commercial allergenic extract or is it commercial raw material from which the authors have made the extract? Do you know the content in the main allergens of Der p? Is it LPS-free?

Ans: Thank you very much. Commercial raw material was used in this study, and we extracted it to obtain Der p extract. The extraction method has been supplemented in the text as follow:

Der p powder was extracted using diethyl ether and then the filtrate was evaporated. PBS was added in to dissolve and transfer to dialysis membrane (cut off point: 12-14 KDa), and then it was dialyzed with water. The dialysate was freeze-dried to powder to obtain Der p extract. The extract was stored at -80℃ before using.

Line 251. The intranasal instillation was a total of 50 µg (50 µl at 1.0 mg/mL), did the mice resist this high concentration of allergens? In our experiments, using individualized allergens 5 µg/instillation is enough.

Ans: Thank you very much. Our method was according our previous study. In our experience, this method could successfully induce allergic asthma. Der p extract used in this article was not purified Der p such as biotinylated Der p protein or recombinant Der p protein. The similar doses were used in other literature as follow.

Kao, S.T.; Wang, S.D.; Wang, J.Y.; Yu, C.K.; Lei, H.Y. The effect of Chinese herbal medicine, xiao-qing-long tang (XQLT), on allergen-induced bronchial inflammation in mite-sensitized mice. Allergy 2000, 55, 1127-1133.

Chu, P.Y.; Sun, H.L.; Ko, J.L.; Ku, M.S.; Lin, L.J.; Lee, Y.T.; Liao, P.F.; Pan, H.H.; Lu, H.L.; Lue, K.H. Oral fungal immunomodulatory protein-Flammulina velutipes has influence on pulmonary inflammatory process and potential treatment for allergic airway disease: A mouse model. J Microbiol Immunol Infect 2017, 50, 297-306.

Line 257. Figure 5. On the line says “Der p sensitized two weeks ago” Is this correct? The text describes that the mice were sensitized by subcutaneous injection on day 0 and 7.

Ans: Thanks for your friendly reminder. The mistake has been corrected.

Line 278-279. Were the cytokines-ELISA used commercial or homemade? If they were commercial please, indicate.

Ans: Thank you very much. The cytokines-ELISA Kits were used commercial, and these were corrected as follow:

The content of various cytokines in BALF, including IL-4, IL-5, Il-10, IL-12, IL-13, and IFN-γ were determined using commercial mouse-specific ELISA kits (BioLegend, San Diego, CA, USA).

Line 291. Statistical analysis.

Did the authors used any statistical software? Please, indicate.

Ans: Thank you very much. The revisions have been added into the text as follow:

Statistical analyses were performed with SPSS software for Windows, version 25 (IBM, Chicago, IL, USA).

The Scheffe’s test running after ANOVA is very flexible and with a low statistical power. It is better to use the Tukey’s test because it have a narrowed confidence interval and a high statistical power.

Ans: Thank you very much. We did Tukey's test according to your suggestion. The results were consistent with Scheffe's test, so we still used Scheffe's test in the study.

RESULTS.

Figure 3. Is the black arrow in the correct position?

Ans: Thank you very much. The mistakes have been corrected.

Table 1. Do the authors have standardized measures of injury severity? How did the authors perform the statistical analysis in this case?

Ans: Thank you very much.

(1) Degree of lesions were graded from one to five depending on severity: 1 = minimal (< 1%); 2 = slight (1–25%); 3 = moderate (26–50%); 4 =moderate/severe (51–75%); 5 = severe/high (76–100%).

(2) Histopathological analyses were performed by the non-parametric Kruskal-Wallis test followed by the Mann-Whitney U-test. These revisions have been added into the text.

Figure 5. Why did the authors not used flow cytometry? It is usually less variable than cell count by hemocytometry and allows specific cell type counts such as eosinophils.

Ans: Thank you very much. Due to the limitations of equipment, we could not use flow cytometry to measure. We used the results of tissue sections and hemocytometry to observe the effect of imperatorin on Der p-induced asthma and inflammation.

Figure 6. (d, e and f) Was the control group (-,-) include in the statistical study? Were the values of the control group (-,-) considered as the cytokine-basal values?

Ans: Thank you very much. The control group was included in the statistical study, and the values of the control group were considered as the cytokine-basal values. We used “#” to show the results of Der p–induced group compared with the control group and “*” to show the results of imperatorin groups compared with the Der p–induced group.

Discussion

Line 171-172. How did the authors count the macrophages?

Ans: Thanks for your friendly reminder. In this study, we didn’t count the macrophages, so the text was revised as follow:

Compared with the sensitized group, the BALF of imperatorin-treated mice exhibited decreased total inflammatory cell count.

Line 186 and 216. Ref. 15 and 20. The authors could compare these results in more detail.

Ans: Thanks for your friendly reminder. It has been revised as follow:

These findings support those of other studies which indicated imperatorin may inhibit may inhibit the degranulation of IgE-mediated mast cells and the histamine release.

These findings are consistent with that of a previous study reporting that the imperatorin can reduce the levels of IL-4 and IL-13 and furthermore inhibit the promotion of IgE.

Conclusions

 Line 299. “IgG1a” must be IgG1.

Ans: Thanks for your friendly reminder. The mistake has been corrected.

Introduction

Line 49-50. “In Taiwan, allergic asthma is primarily caused by dust mites”. The sentence needs a reference.

Ans: Thanks for your friendly reminder. The literature has been cited in the manuscript.

Lai, C.L.; Shyur, S.D.; Wu, C.Y.; Chang, C.L.; Chu, S.H. Specific IgE to 5 different major house dust mites among asthmatic children. Acta Paediatr Taiwan 2002, 43, 265-270.

Line 62. Reference [2] appears after reference [3]. They are unordered.

Ans: Thanks for your friendly reminder. The typesetting of the literatures in the manuscript was performed using the Endnote software. The reference [2] was existed in Line 49, and reference [3] was shown later.

Line 77. “Mill” would be in italic.

Ans: Thanks for your friendly reminder. "Mill" is the authority name, so we used the roman font in the article. However, to be consistent with the scientific name written earlier in the article, we have deleted this word.

Line 77-78. The sentence needs references to support the beneficial effect of imperatorin. It would be convenient to indicate the existence of toxic effects of this compound (i. e. Sondhia et al., 2012).

Ans: Thanks for your friendly reminder. The references have been cited in the paper and the toxicity of imperatorin has been stated as follow.

It exhibits several pharmacological properties including anticancer, anti-inflammatory, antioxidant, neuroprotective, and immunomodulatory effects, but it might have hepatotoxicity in mice at higher dose (40 mg/kg).

Round 2

Reviewer 2 Report

The article has improved significantly. However, there are still typographical errors that the authors should correct:

Line 158: Figure 6. "EEfects" must be corrected.
Line 193: "may inhibit" is duplicated.